# Revealing undergraduate biology students' conception of variability and error bars within graphing

Lauren Stoczynski[1], David Zis[2], Anna Woodruff[2], Susan Maruca[3], Eli Meir[3], Joel K. Abraham[4], Stephanie M. Gardner[2]*

**1** Department of Biological and Environmental Sciences, Le Moyne College, Syracuse, New York, United States of America, **2** Department of Biological Sciences, Purdue University, West Lafayette, Indiana, United States of America, **3** Simbiotic Software Inc. Research Division, Missoula, Montana, United States of America, **4** Department of Biological Sciences, California State University at Fullerton, Fullerton, California, United States of America

* sgardne@purdue.edu

## Abstract

Creating and interpreting graphs are quantitative reasoning skills all science, technology, engineering, and math students need to develop. In biology, data are lacking in our understanding of how undergraduate students interpret variability across different graph types or what knowledge students have about the data error bars represent. This study analyzed 3506 student responses within a graphing assessment. We developed a code book for students' open-ended responses on the data error bars represent and used chi-square tests to look for significant differences in how students described error bars based on a graph they created and self-reported demographic data. We observed four major categories in how students described error bars: broad terms, error terms, purpose terms, and trend & analysis terms. When responses were linked with a graph the student made, those who created a bar graph with a bar for each data point showed lower understanding of error bars compared to students who themselves created a bar graph with aggregated means and error bars. We also observed differences in how students described error bars based on their major but did not observe differences based on whether students were in introductory or upper-level courses. Our results suggest that students are not receiving enough instruction on variability within graphing even though bar graphs with error bars are a common graph that students are asked to construct. More scaffolded repeated instruction is needed across biology curricula.

## Introduction

Quantitative reasoning, the application of math and statistical skills [1], is important for Science, Technology, Engineering, and Math (STEM) students' development within their discipline. These skills are integrated into the core competencies of the

**Data availability statement:** All relevant data are within the manuscript and its Supporting information files.

**Funding:** This research received grants from the National Science Foundation under grant Nos 1726180 and 2111150. Any opinions, findings, and conclusions or recommendations expressed in this material are those of the author(s) and do not necessarily reflect the views of the National Science Foundation. The funders had no role in study design, data collection and analysis, decision to publish, or preparation of the manuscript.

**Competing interests:** The authors have declared that no competing interests exist.

AAAS initiative Vision and Change [2] and are centered in other works breaking down important curriculum outcomes for Life Science faculty [3,4]. The Bologna Process indirectly involves incorporating quantitative reasoning skills within curriculum reform in European higher education [5]. With biology being increasingly data centric, students should receive more instruction in their biology courses teaching the necessary statistical knowledge in a biological context [6].

The display of data in graphs is important within STEM generally and within biology. When creating graphs, students should grapple with messy data, which requires multiple quantitative skills, including statistical thinking [7,8]. The Graph Construction Conceptual Model-Biology (GCCM-Bio) provides a framework for this instruction with four components: Data Selection, Data Exploration, Graph Assembly, and Graph Reflection [9]. Statistics and data variability fall within the Data Exploration component, with other quantitative skills such as identifying data type, scaling axes, and finding and interpreting trends in other components. The basic competencies of biological experimentation include identify, analyze, communicate, question, conduct, plan, and conclude [10]. Within the plan competency, students should think about variation regarding whether the system has inherent natural variation. Students should differentiate between this natural variation and experimental variation. Constructing appropriate data visualization for an experiment is part of the analyze competency and there may be a disconnect for students between the planning and analysis of their experiments.

Students' ability to interpret variability within a graph falls within the essential quantitative reasoning skills STEM students should learn. Research studies that investigate student understanding of variability in a graphing context within statistics courses provide some useful insights. Chaphalkar & Wu [11] observed that over a semester, a statistics course did not increase students' understanding of variability shown in histograms and dot plots. Students still showed different levels of reasoning including being unable to meaningfully discuss variability between two graphs, recognizing one relevant part of the graph when explaining variability, and using several statements to show a more integral understanding of variability. When delMas & Liu, Yan [12] had elementary students interpret standard deviations between different histograms, they noted that some students had rudimentary notions of standard deviation, using terms such as range, mean in the middle, and far away values as justifications for one standard deviation being larger or more correct terminology including far away mean, equally spread out, and bell-shaped justifications. The teaching of statistics and statistical concepts in the life sciences is complex. A meta-analysis looking at life science programs in Canada found that around 20% of programs do not require a statistics course and roughly 45% require one statistics course. Statistics courses generally separate biology content from statistical content resulting in students struggling to integrate statistical and biological knowledge [13]. While statistics courses are an important part of undergraduate biology curriculum, general statistics concepts including variability should be included across biology courses, where repeated practice can lead to an increase in these critical thinking skills [14].

Students' knowledge of variability within graphing has been analyzed extensively using histograms and dot plots [12,15–18]. When analyzing histograms and value bar

charts (a bar graph where the top of the bar represents an individual data point instead of a mean), students who did not consider the graph type may confuse the interpretation of greater variability with the height of the bars. Students also incorrectly thought "greater variability of bar heights indicates greater variability in the data" [15]. Because students are asked to construct bar graphs with averages, line graphs, and scatterplots in biology, we lack an understanding of how students interpret variability within more applicable graph types.

While a tool exists to measure incoming statistical knowledge of introductory biology students [19], this instrument does not ask questions directly dealing with variability in data or graphs. Other instruments have been developed to understand variation in biological experimentation with less emphasis on variability in graphing [20], and with few questions dedicated to directly measuring students' understanding of variability within graphs [21]. Evidence has shown that introductory biology courses should include more practice for the development of quantitative reasoning skills, especially data visualization [22]. Biology instructors may have bias by assuming students already know how to read basic graphs, and not until having students again in upper-level classes did instructors realize that quantitative reasoning skills would be important to include within introductory courses [23]. Evidence shows implicit teaching of quantitative reasoning skills is not enough for students to make progress in developing these necessary skills [24], thus we need to understand more fully what students know about variability in graphing so that instruction can be explicitly developed and implemented into biology courses.

Variability can be depicted in several ways. For bar graphs, which typically show means across groups, error bars show variability. Variability in error bars can represent confidence intervals, standard deviation, standard error, range, or other measurements. While bar graphs with error bars are still used ubiquitously throughout the literature, in many cases the error bars may be unlabeled [25]. Descriptive error bars show the spread of the data compared to inferential error bars that allow for comparisons between groups [26]. One common misconception of error bars is the "all or nothing" interpretation where values from the distribution of the data are either found within the error bar or not [25]. Even practiced researchers have trouble interpreting error bars; in one study, 31.5% of peer reviewed authors surveyed misinterpreted error bars showing confidence intervals or standard error for significance in a graph. The results also showed that researchers failed to describe the differences between the information conveyed between error bars that show confidence or standard error [27]. In the limited literature on how biology undergraduate students use error bars, undergraduates graphed either raw points or just averages, while only a few graphed averages and added error bars compared to graduate students and professors who tended to always graph averages with error bars [28]. Picone et al. [29] included many exercises in environmental science and ecology courses to increase undergraduate students' ability to interpret ecology graphs. The researchers increased student proficiency in interpreting graphs and constructing proper scatterplots and bar graphs. However, one of the areas that students continued to struggle, despite a semester of activities geared toward making and interpreting graphs, was discerning general trends amid variability within data. Our own research has revealed that students rarely plot aggregate data and when they do they don't include a measure of variability like error bars [28,30–32]. We know little about how undergraduate biology students interpret variability specifically within different graph types and what data biology students think error bars represent within a graph.

Our research team developed a performance-based digital assessment of graph construction competence (GraphSmarts) embedded in six biological scenarios that require students to make and interpret graphs [9,31,33]. Within the context of the data collected through students completing GraphSmarts scenarios, we have the following research questions:

1. How do students identify variability within a treatment across different graph types?

2. How do students describe the data that error bars can convey on a bar graph?

3. How does the graph a student constructs lead to insight on a student's understanding of variability in a graphing context?

4. How does a student's year of undergraduate study, major, and the course they were completing impact their responses and graphs created?

## Methods

### GraphSmarts assessment

This study was part of a larger project called GraphSmarts, designed to improve understanding of undergraduate students' competence in graph construction. We have six graphing assessments, based on the skills outlined in the GCCM, covering a variety of biology subdisciplines [33]. Within each assessment students were given some general background information, a hypothesis, and three predictions. For each prediction, students were asked to create a graph within our constrained graphing software and make claims about whether the graph supports or refutes the prediction. At the end of each assessment there were two questions which focused on knowledge of variability. First, a dropdown question (Fig 1), asked "how easy or hard is it to interpret the amount of variation within the categories" across six different graphs. We recognize that the wording from this response incorrectly uses the term variation instead of variability, however, due to the general interchangeability of variation and variability we do not think this impacted student responses based on student interviews. We will use the term variability through the rest of the text. The possible responses were 'easy to interpret', 'hard to interpret', 'variability not shown', or 'I am not sure'. The graphs all show the same hypothetical categorical variable on the x-axis and hypothetical quantitative variable on the y-axis. Three graphs show raw data, and three graphs show means. Our "correct" answers are shown in Fig 2, where graphs that show variability as error bars or show all points in an orderly manner were "easy" to read. The second question (Fig 1) showed a bar graph with error bars using data from that particular scenario. An open-ended question asked "What types of information do the error bars provide you about the data represented in each bar? If you are unsure, please respond with 'I don't know'". This question was purposefully left broad and the type of error bar on the graph was not provided to see how students would respond.

### Student population

This study was conducted under IRB approval with protocol CSU-Fullerton, #HSR-20-21-357. Students gave electronic consent to have their data used for research purposes. Instructors were recruited to administer the assessments to their students as an assignment within their course between August 1, 2022, and May 31, 2024. Students represented over 70 classes from over 40 institutions across the United States. Students from R1 institutions made up 70.2% of our data followed by R3 institutions at 9.2%, R2 institutions at 7.2%, Masters/PUI institutions at 7.1%, and Tech/Community Colleges at 5.9%. If a student completed more than one assessment, data from the first or only time was included in this study (N = 3506). Students reported demographic information at the end of the assessment (Table 1). For majors, a student could self-report as a general biology major. Specialized biology majors were those students who reported studying either cellular/molecular biology, ecology and evolution, neurobiology, or immunology. Examples of allied majors include non-biology STEM majors such as engineering or chemistry students.

### Disaggregating the data

We chose to disaggregate the data to investigate more specific questions in how students understand variability and error bars. To answer question three, we linked student answers with data on their constructed graphs for one of the three predictions in each assessment. The correct graph for the prediction chosen would use a categorical variable on the x-axis and a quantitative variable on the y-axis, to provide the option to create a graph similar to the one seen when the students answered the open-ended question about error bars.

### Code book development

We used inductive and deductive coding and thematic analysis to analyze and describe our qualitative data from student responses [34–37]. The codebook was generated after an initial phase of analytic memoing that was inductive in nature. A subset of student responses was memoed leading to a set of codes that emerged from the

 

Q1.11 Below are six graphs of hypothetical data, all of which display the same data set. Some graphs show field data (blue) while others show means for categories A, B, and C (yellow). For simplicity, axes are not labeled.

For each graph, how easy or hard is it to interpret the amount of variation within the categories (A, B, C)? (Circle an answer below each graph)

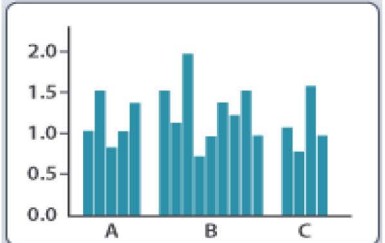

O  Variation easy to interpret
O  Variation hard to interpret
O  Variation not shown
O  I'm not sure

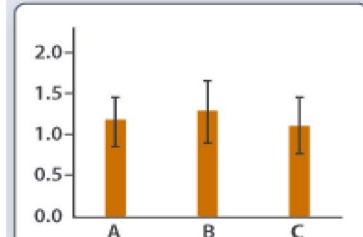

O  Variation easy to interpret
O  Variation hard to interpret
O  Variation not shown
O  I'm not sure

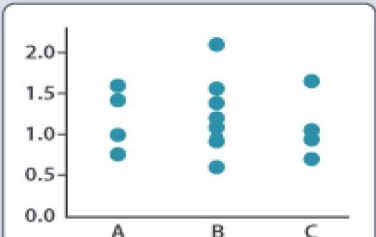

O  Variation easy to interpret
O  Variation hard to interpret
O  Variation not shown
O  I'm not sure

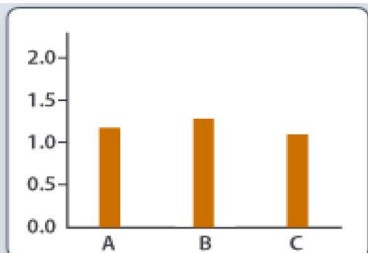

O  Variation easy to interpret
O  Variation hard to interpret
O  Variation not shown
O  I'm not sure

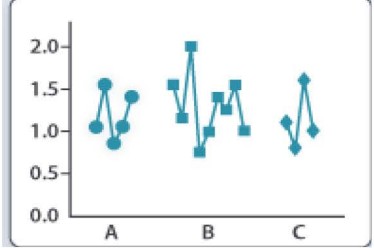

O  Variation easy to interpret
O  Variation hard to interpret
O  Variation not shown
O  I'm not sure

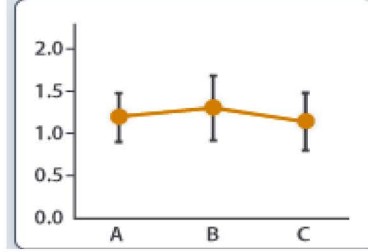

O  Variation easy to interpret
O  Variation hard to interpret
O  Variation not shown
O  I'm not sure

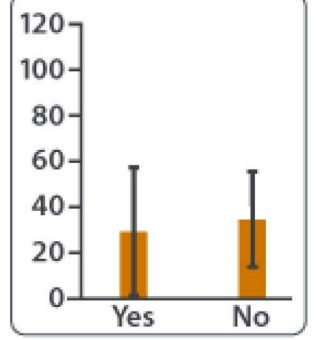

Q1.12 The image above shows a bar graph with error bars. What type of information do the error bars provide you about the data represented in each bar? If you are unsure, please respond with "I do not know".

**Fig 1. Example of the questions students answered from the assessment.** The dropdown is the exact same for each scenario, Q.12 shows a bar graph with relevant X axis variables for that scenario.

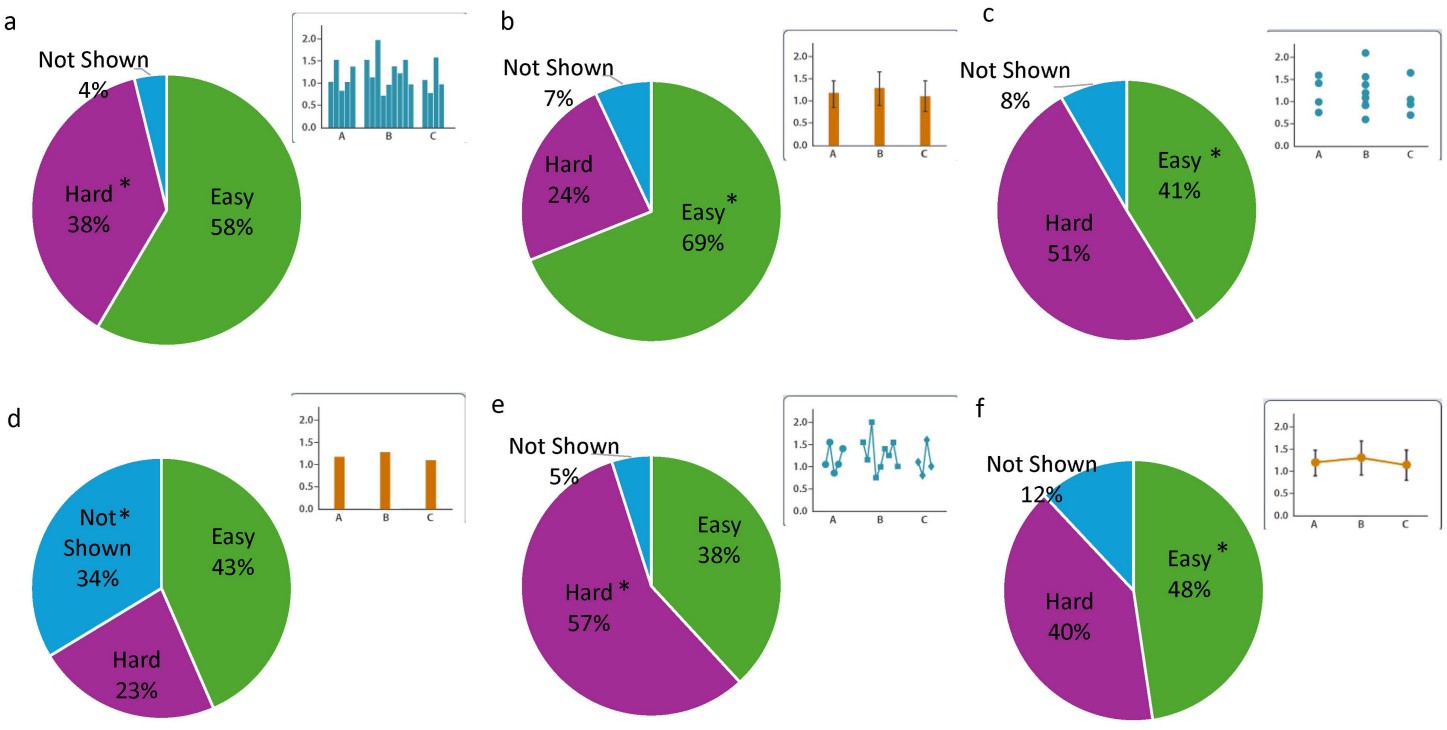

**Fig 2. Aggregated data on how students answered dropdown questions.** Asterisk denotes the correct answer.

data. After this inductive phase, we compared the codes to those already described in the literature. This resulted in a set of codes consisting of deductive codes that fit with terminology already used to describe error bars by biologists [26] and inductive codes to capture all of the student responses [34–37]. Two undergraduate students were brought onto the project who helped with iterative coding using several new subsets of student responses. We refined code definitions, incorporated new codes, and organized the codes into the four current categories. A third undergraduate was brought on to the project and was trained on the codebook using several subsets of student responses.

After the codebook was finalized and described our data, LS coded 100% of student responses and 20% were split between and independently coded by two undergraduate researchers [38]. The undergraduates and LS then discussed all codes in that 20% sample. After completing code checking sessions, inter-coder similarity was evaluated in two ways. If after code checking LS initial codes were upheld, inter-coder reliability was 91.5%. Overall, when accounting for both upheld and changed codes from LS, the inter-coder similarity was 81.3%.

## Quantitative data analysis

All data analysis was conducted in R 4.3.1 at a significance level of 0.05 [39]. We wanted to know whether code frequencies, dropdown answers, and answer confidence were associated with other variables such as graphs made, student year, major, or course type. A chi square test of independence was conducted using the chisq.test() function. We used a fisher's exact test with the fisher.test() function when the sample size was below five in any given category within a contingency table as this test is more accurate with smaller sample sizes. To look for significance in the number and types of codes used when explaining error bars, a Kruskal Wallis test was run with the kruskal.test() function. When a significant result was obtained, a Dunn test was used to assess which comparisons were significant using the dunnTest() function in

**Table 1. Demographic data of aggregated data from student assessments (N = 3506), students could choose not to disclose this information.**

| Demographic | | Frequency |
|---|---|---|
| Course Type | Introductory | 0.85 |
| | Upper-Level | 0.09 |
| | Nonmajors | 0.06 |
| Year | First year | 0.4 |
| | Sophomore | 0.21 |
| | Junior | 0.08 |
| | Senior | 0.09 |
| Major | General Biology (GenBio) | 0.33 |
| | Specialized Biology (SpecBio) | 0.21 |
| | Allied | 0.23 |
| | nonSTEM | 0.09 |
| Gender | Woman | 0.46 |
| | Man | 0.21 |
| | Trans | 0.01 |
| Ethnicity | Native American | 0.001 |
| | Asian | 0.13 |
| | Black/African American | 0.05 |
| | Hispanic | 0.05 |
| | White | 0.35 |
| SocioEconomic | First Generation | 0.3 |
| | Pell Grant Recipient | 0.13 |
| | International Student | 0.04 |

the FSA package [40]. All analyses used a Bonferroni correction to reduce the probability of Type I error. Critical values and residuals from chi-square tests can be found in Appendix A.

## Results

### Many students had difficulty recognizing depictions of variability

Responses to the dropdown question varied compared to our correct responses (Fig 2). We cannot guarantee that some students were interpreting variability across treatments instead of within treatments. Over half of students thought the raw bar graph variability was easy to interpret (Fig 2A) and the raw categorical scatterplot was hard to interpret (Fig 2C). Only 34% of students correctly said that the bar graph with just means did not show variability (Fig 2D). In agreement with our correct responses, students thought the line graph with the raw data was hard to read (Fig 2E). Roughly 70% of students recognized that the bar graph with error bars was easy to interpret (Fig 2B). There was more of a split in how students interpreted the line graph with error bars (Fig 2F).

### Student descriptions of error bars fell into four categories

We used inductive coding to analyze student responses to the question "What types of information do the error bars provide you about the data represented in each bar? If you are unsure, please respond with 'I don't know'". Our codes fell into four categories: *Broad Terms, Error, Purpose,* and *Trend & Analysis* (Table 2). Student responses could fall within multiple categories. *Purpose* codes were used the most, followed by *Broad Terms, Trend & Analysis,* and *Error* (Fig 3a). Additional examples beyond what is shown in the results can be seen in our code book (Appendix B). We understand that

**Table 2. Definitions for four categories observed from analyzing student responses to what information error bars represent.**

| Category | Definition | Codes in That Category |
|---|---|---|
| Broad Terms | Vague definitions used to describe data in error bars | Variability (alone)<br>Variability in data<br>Uncertainty (alone)<br>Uncertainty in data<br>Significance |
| Error Terms | Different terms related to error that the error bars may represent | Error<br>Error in data<br>Margin of error<br>Error in experimentation<br>Accuracy and precision |
| Purpose | Terms relating to the type of error bar of a purpose for using the error bar | Standard deviation<br>Standard error<br>Confidence interval<br>Outliers<br>Distribution<br>Significance testing<br>Lowest and Highest points/Range |
| Trend & Analysis | Terms for ways the error bars could be used to analyze trends in the graph | Mean/Average<br>Comparison<br>Distance from the mean<br>Size<br>Overlap<br>Statistical test |

the data we have may not show full student knowledge on the content of error bars. However, we are confident that our large dataset does capture interesting trends that are meaningful.

### Broad term codes

Codes within the *Broad Terms* category (24%) encompassed vague phrases that talk about what data are represented by the error bars. Two common codes within this category discussed the variability and uncertainty that error bars represent. Students most commonly referred to the variability in the data (47%) followed by the uncertainty in the data (34%). Students also used the terms variability (6%) or uncertainty (5%) without providing context (Fig 3b). A sophomore student from an introductory course who was majoring in a specific biology field wrote, "The error bar allow[s] us to interpret variation and start to see if data is significant." This student also used significance, but without purposeful reference to the data, which we describe as a broad term.

### Error codes

Codes in the *Error* category include several phrases representing different types of error (Fig 3c). A first-year R1 student in an introductory course majoring in a specific biology field wrote, "The error bars are there to show uncertainty or any **error in a data set** when graphing." More specifically, students may have talked about error in collecting the data, human error, or error in calculations as examples of error in experimentation. An R1 first-year student in an introductory course in an allied major stated, "The data provided from error bars would be how there is some type of uncertainty [that] could emerge [from] **errors accidentally collected during measurements**." Additionally, students also used the phrase 'margin of error' in their discussion of error bars, such as this R1 first-year student in an introductory course majoring in a specialized biology field, "The point of error bars on a graph are to indicate the error or uncertainty of a certain measurement (shows reader there might be a certain **margin of error** up to a certain point in the data)." Similar to variability and uncertainty,

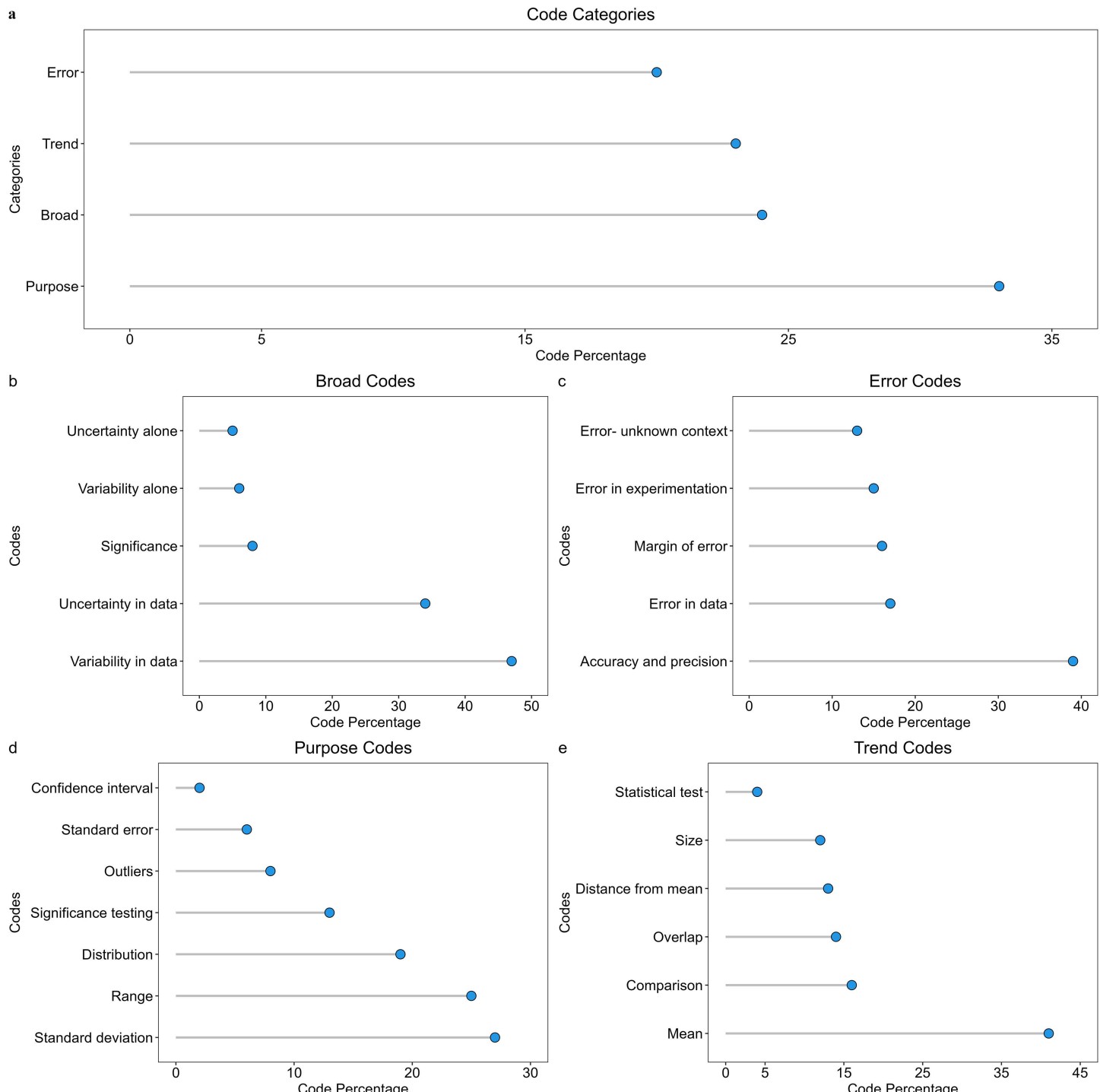

**Fig 3. Frequencies of codes in how students responded to the question, "What type of information do the error bars provide you about the data represented in each bar".** (a) shows frequencies out of all students (N = 3506), frequencies do not add up to 100 because students could have used codes from multiple categories. Within each category are the percentage of students who used codes for (b) Broad Terms, (c) Form of Error, (d) Purpose, and (e) Trend & Analysis. The number of codes used to calculate frequencies are 1433 for Broad Terms, 1082 for Form of Error, 2097 for Purpose, and 1400 for Trend & Analysis.

some students discussed error in a way that could have multiple meanings. The most commonly used code in this category is when students describe error bars as informing on the accuracy or precision of the data or experiment (39%). Some students discuss accuracy or precision in a way that is similar to chemistry or physics principles of how accurate or precise the data should be, such as this R1 first-year student from an introductory course majoring in general biology, "The error bars show how **accurate** the data is. The smaller the error bar the more **accurate** the data is." Other students discuss accuracy or precision by describing the error bars as showing how far the true value is from the reported data, as shown by this R1 first-year student from an introductory course in an allied major:

> "Error bars are graphical representations of the variability of data and used on graphs to indicate the error or uncertainty in a reported measurement. Error bars give a general idea of how precise a measurement is or conversely **how far from the reported value the true the value might be**."

### Purpose codes

The *Purpose* codes highlight some of the different types of error bars and their uses ([Fig 3d]). Students referred to the different types of error bars as standard deviation (27%), standard error (6%), and/or confidence intervals (2%). Many students would list one, but students may also list many such as this first year allied major at a community college from an introductory course:

> "Because the error bars are not identified as representing the **standard deviation, standard error, or a confidence interval**, the information they provide is unclear. However, in general, one could assume that true lobster density for each category is contained somewhere within the range shown by the error bars."

Other uses for error bars were noted by students such as their ability to detect a range (25%), the distribution of the data (19%), and/or be used for significance testing (13%). Here is how a first year R1 student majoring in a specialized biology field taking an introductory course describes error bars showing the distribution in the data: "Since the bars are averages for the groups, the error bars **show the high and low points of the data** that are not conveyed by the bars themselves." Students may also incorrectly say the error bars represent outliers (8%) within the data. One such example from an R1 sophomore taking an introductory course and majoring in general biology, "The error bars **show the outliers in the data** and the variation in the sample size. It also tells the interpreter whether the data is statistically significant or not."

### Trend & analysis

*Trend & Analysis* had students recognizing the graph showed aggregated data in the mean/average (41%). Students could also make a statement about the graph that would lead to a conclusion about the data ([Fig 3e]). A response received a comparison code (16%) when the student discussed comparing the treatments in the graph without mentioning overlapping error bars, such as this senior R2 student in an upper-level course majoring in general biology, "Error bars represent the variability of the data and how accurate the measurement is. In this case, **NoFishing habitat had more spread in the data when compared to YesFishing habitat**." Some students discussed the size of the error bars (12%), such as this first year PUI student in an introductory course majoring in general biology, "The error bars show the standard error of the mean and show the amount of uncertainty. **Shorter error bars** are more certain and **longer error bars** are less certain." Discussion of overlapping error bars (14%) was common with the significance testing code as students recognized that the overlap in the error bars could help them make a conclusion about the data, such as this senior allied major at a masters institution student from a nonmajor course:

"The type of information an error bar provides is how statistically significant data is. For this data the error bar show[s] a lot of **overlap** between both groups. A lot of **overlap** shows that **the difference is not statistically significant**. Therefore further analysis should be conducted to identify trends and formulate a conclusion."

Some students recognized that the error bars could represent a statistical test (4%), which we also grouped with the recognition that the true mean could fall within the error bars, such as this senior allied major from an R3 institution taking a nonmajor course:

"The graph's error bars indicate the variability or precision of the data. If these are standard errors, they illustrate how much the sample mean of survival time for patients with and without p53 mutations **is likely to differ from the true population mean**. This explains how confident we can be in the data represented by the bars."

Around 13% of students who had a *Trend & Analysis* code for their response talked about error bars representing some distance from the mean. This first-year R1 introductory course student in an allied major refers to the distance as a range, "The error bars show the standard deviation from the mean which is shown in yellow. This shows roughly a **range from the mean** that the data had." Other terms for distance included 'deviation' and 'how far away', which are seen from this first-year community college student majoring in general biology in an introductory course, "This displays that the average of each is where the orange bar ends. The error bars are the **deviations away from that mean** for the data set," and this sophomore R2 student in an introductory course majoring in general biology, "The information that the error bars show the amount of uncertainty in the data. The bar graph shows the mean/average and then the error bars typically show how far away the data points are **away from the mean**."

**There was no one predominant combination of categories in student answers**

There were 15 combinations of how students could talk about error bars using our four main categories (the 16th combination would be none which is the "I don't know" code). The proportion of students who used these combinations can be seen in Fig 4. Students explained the meaning of error bars the most using only *Purpose* codes (14.1%), followed by *Purpose* and *Trend* codes (11.6%), only *Broad* (10.1%), and only *Error* (8.6%). Here is an example of *Purpose* and *Trend* combination by a senior at a masters institution in an upper- level course majoring in general biology:

"The error bars tell us how big of a spread our data spans across. The longer error bar on the "No" bar tells us that that data is more spread out than the "Yes" data and has a wider range."

Combinations seen the least include excluding purpose (1.5%), excluding trend (1.5%), and all categories (2.1%). Here is an example of a senior R2 student in an introductory course majoring in general biology who used codes from all four categories to explain what error bars represent:

"The error bars on the bar chart provide information about the variability or distribution of the data for each category ("yes" and "no"). Specifically, error bars typically represent one standard deviation above and below the mean, or can represent other measures of variability. This allows the reader to understand the range and reliability of the data, as well as how much individual data points may differ from the average."

**Most students appeared confident in their interpretation of error bars**

We also measured student confidence in their response. Students received a "yes" if there was no indication that they were unconfident in their response (77%). Students were unsure if they used terminology such as "I believe", "unsure",

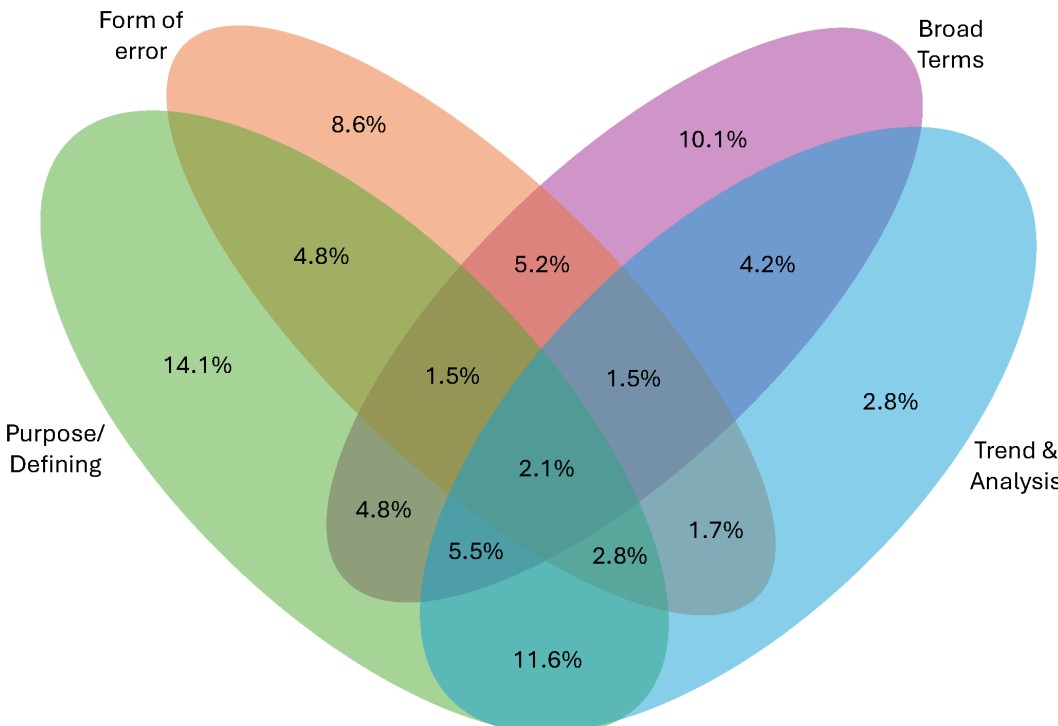

**Fig 4. Venn Diagram showing the breakdown of categories used across student assessment (n = 3506).**

"assume", or using a question mark, which suggested they were unconfident in their response but decided to try and give a response anyway (5%). Students may also use a form of "I don't know, but" in their response such as this first year R1 student in an introductory course majoring in a specialized biology field, "I do not truly know, but I imagine error bars to be similar to cat plot charts, showing the highest value and lowest values of the data to show the range." If students did not know the answer, they were prompted to simply reply, "I don't know" (19%).

### Students found variability within a treatment easier to interpret on graphs similar to the ones they themselves made

We paired a student produced graph with their answers for the dropdown question on interpreting variability within a treatment. We used a chi-square test to determine if there was any relationship between the graph a student made and their ability to recognize variability. We observed significant differences in the frequencies of how students answered the dropdown question based on whether the graph from the dropdown matched the graph the student created (Fig 5). Students who made a raw bar graph were more likely to say the raw bar graph variability was easy to interpret ($X^2 = 142.69$, df = 10, p < 0.0001; Fig 5a), while stating the bar graph with error bars was hard to interpret. For the bar graph with error bars ($X^2 = 37.88$, df = 10, p = 0.0002), students who also made bar graphs with error bars were more likely to say the variability was easy to interpret (Fig 5b), however these students also found the bar graph of raw data harder to interpret. For the raw categorical scatterplot ($X^2 = 34.52$, df = 10, p < 0.0001), students who also made a raw categorical scatterplot were more likely to say the variability was easy to interpret (Fig 5c).

Even though there is some opinion to the dropdown answers, the bar graph just showing means has a definite answer because there is no variability shown if trying to interpret the variability within a treatment. For this graph, students who

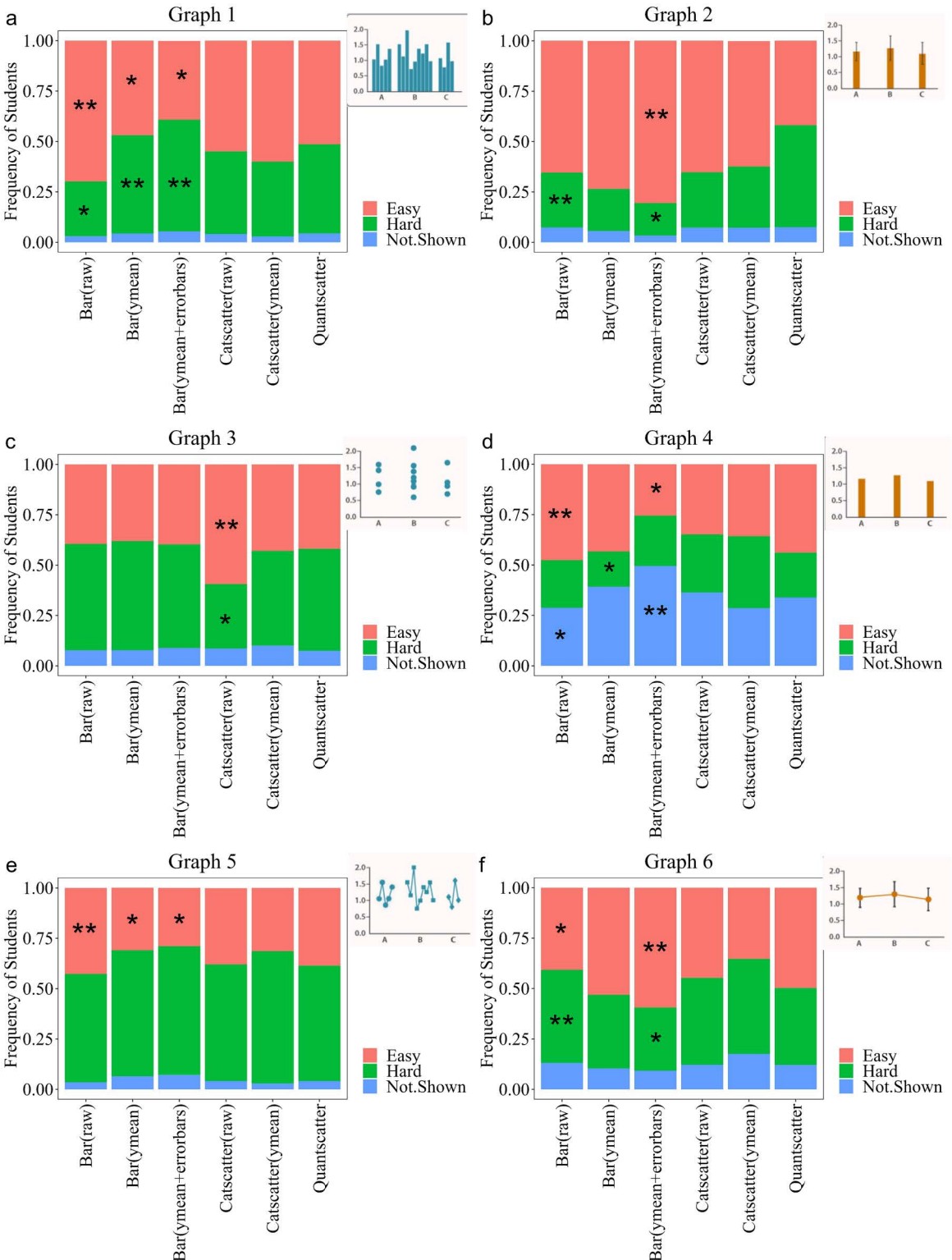

**Fig 5. Frequency of student responses to the dropdown question based on a student created graph within the GraphSmarts assessments.**
The graph being analyzed is in the upper right corner for each stacked bar graph. A chi-square test was performed, and a single asterisk represents a frequency that was significantly lower than expected. Double asterisk represents a frequency that was significantly higher than expected. Bar (raw) is

a bar graph created with raw data; N = 1071. Bar (ymean) is a bar graph with just means; N = 484. Bar (ymean + errorbars) is a bar graph with means and error bars; N = 391. Catscatter (raw) is a graph similar to graph 3; N = 205. Catscatter (ymean) is a graph with a dot representing the mean for each group; N = 73. Quantscatter is a traditional scatterplot with two numerical variables; N = 184.

made a raw bar graph were less likely to identify that the variability was not shown and more likely to say the variability was easy to interpret ($X^2 = 88.25$, df = 10, p < 0.0001). Students who created a bar graph with error bars were more likely to correctly identify that the variability was not shown (Fig 5d).

For graphs in the dropdown question that did not have any matching student-made graphs, trends still existed based on the type of bar graph a student made. For the line graph ($X^2 = 39.81$, df = 10, p = 0.0002), students who made a raw bar graph were more likely to say the variability was easy to interpret, while students who made a bar graph with error bars were less likely to say the variability was easy to interpret (Fig 5e). For the categorical scatterplot with error bars ($X^2 = 49.72$, df = 10, p < 0.0001), students who made raw bar graphs were more likely to say the variability was hard to interpret, while those that made bar graphs with error bars were less likely to say the variability was hard to interpret (Fig 5f).

### The type of bar graph a student created may show evidence of their knowledge of error bars

We were interested in whether the graph a student created could provide insights into their knowledge of what error bars represent. Across the six scenarios, there were five graphs that could have been made to show a categorical variable on the x-axis and a quantitative variable on the y-axis. If students chose to make a bar graph, they could have made a bar graph of raw data (n = 1071), aggregated data just showing the means (n = 484), or showing the means and representing variability by including error bars (n = 391). Students also could have chosen to make a categorical scatterplot with either raw data (n = 205) or showing means (n = 73; not a large enough sample size had means with error bars for the categorical scatterplots). The sixth graph was a quantitative scatterplot (n = 184), where the student would have chosen at least one incorrect variable.

Significant results from chi-square analysis show a continuum between students who make raw bar graphs and those that make bar graphs with error bars (Figs 6 and 7). Within the main categories ($X^2 = 64.75$, df = 15, p < 0.0001), students who made raw bar graphs were less likely to use *Trend & Analysis* codes compared to students who made error bar graphs (Fig 6a). Within the combinations of main categories ($X^2 = 250.42$, df = 70, p < 0.0001), students who made raw bar graphs were more likely to use only *Error Codes*, while students who made error bar graphs were more likely to exclude *Error Codes* and use *Trend & Analysis* codes in their responses (Fig 6b). Even though students who made an error bar graph were less likely to use error codes, if they did use error codes, they were more likely to use accuracy and precision in their responses ($X^2 = 48.73$, df = 20, p = 0.0002). Students who made raw bar graphs were less likely to use accuracy and precision codes (Fig 6d). Within *Purpose* codes ($X^2 = 61.69$, df = 30, p = 0.0007), students who made a raw bar graph were more likely to discuss error bars representing outliers in contrast to students who made error bar graphs (Fig 6e).

The number of codes used differed depending on the graph made (Fig 6). In general, the number of codes used to talk about error bars increased from students who made a raw bar graph, to students who just graphed the mean, to students who also graphed with error bars. This trend was observed for the main categories (249.23, df = 5, p-value < 0.0001), *Broad Terms* (H = 50.24, df = 5, p-value < 0.0001), *Purpose* (H = 138.9, df = 5, p-value < 0.0001), and *Trend & Analysis* (H = 195.7, df = 5, p-value < 0.0001; Fig 6A, 6B, 6E and 6F). We selected several quotes to show differences in how students described error bars based on the graph type they created (Table 3).

### Graphing ability matches with confidence in describing error bars

Answer confidence varied depending on the graph a student made ($X^2 = 102.49$, df = 10, p < 0.0001), their year ($X^2 = 21.5$, df = 6, p = 0.001), or major ($X^2 = 107.18$, df = 6, p < 0.0001; Fig 8). Students who made a raw bar graph were more likely to

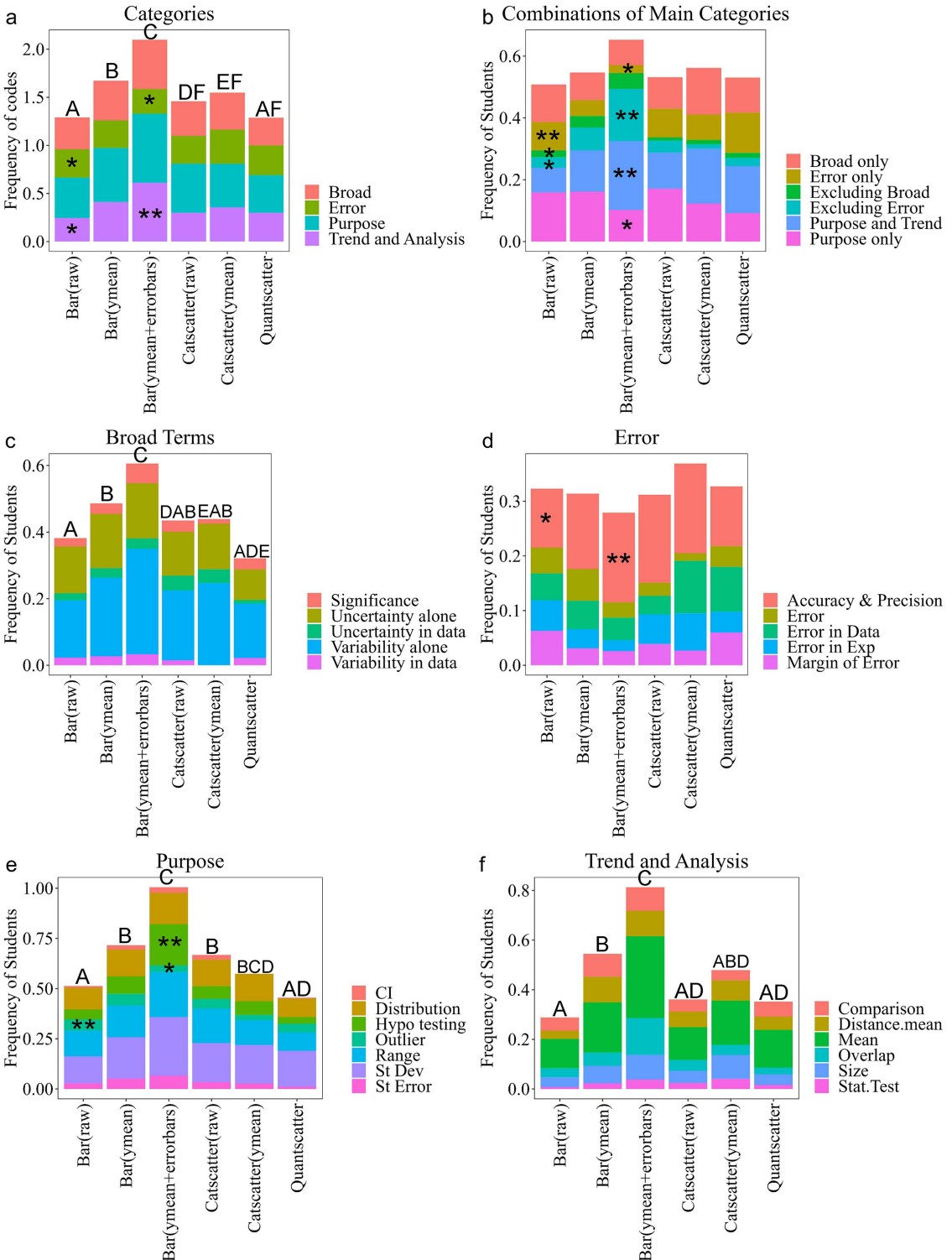

**Fig 6. Frequency of codes used from student explanation of data error bars represent across the categories (a), combinations of categories (b), and specific codes within the categories (c, d, e, f) based on the graph a student created within the GraphSmarts Assessment.** A chi-square test was performed and a single asterisk represents a frequency that was significantly lower than expected. Double asterisk represents a frequency that was significantly higher than expected. Letters above bars represent significant differences from post hoc Dunn test after a significant Kruskal Wallis test.

Bar (raw) is a bar graph created with raw data; N = 1071. Bar (ymean) is a bar graph with just means; N = 484. Bar (ymean + errorbars) is a bar graph with means and error bars; N = 391. Catscatter (raw) is a graph similar to graph 3; N = 205. Catscatter (ymean) is a graph with a dot representing the mean for each group; N = 73. Quantscatter is a traditional scatterplot with two numerical variables; N = 184.

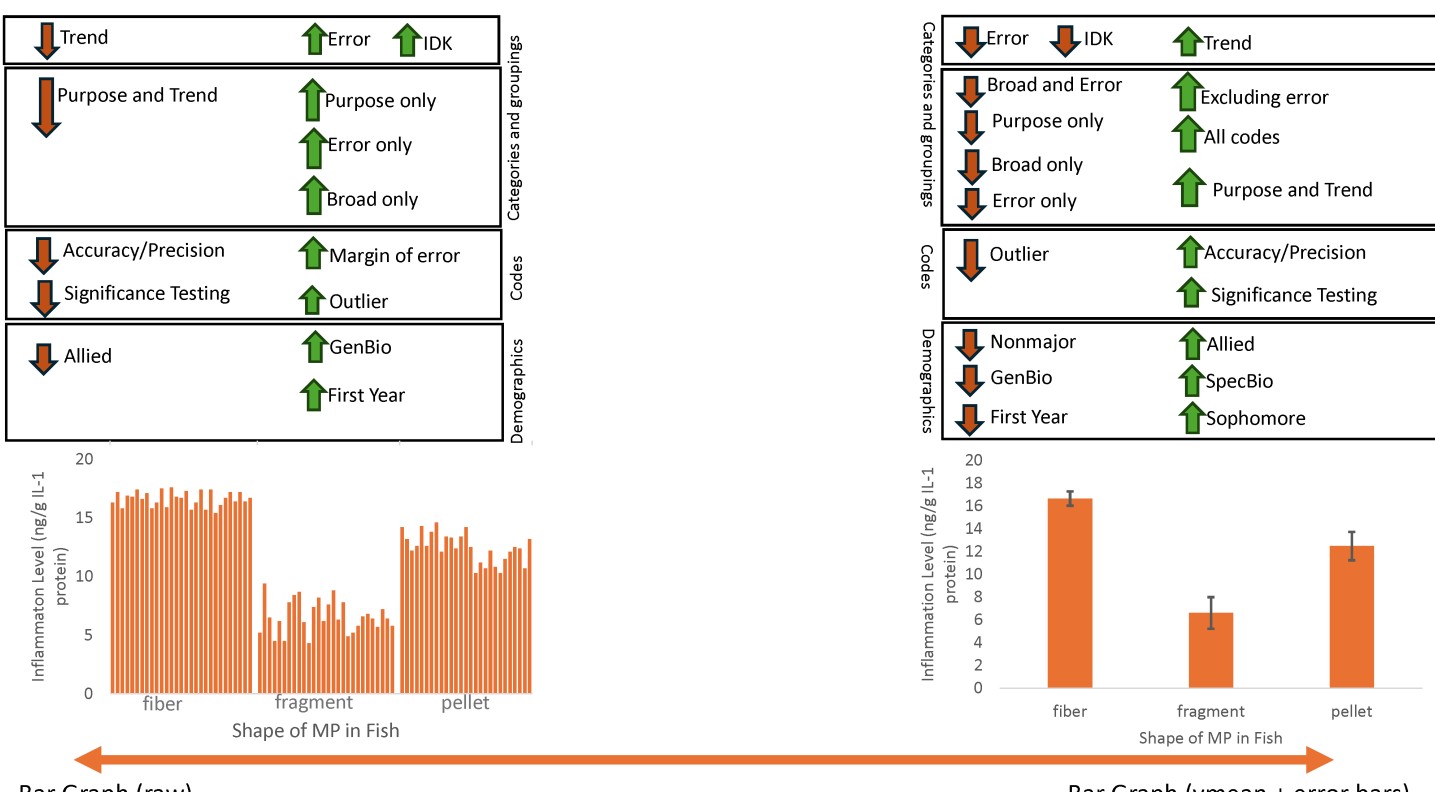

**Fig 7. Graphic showing the continuum results between raw bar graphs and bar graphs with error bars.**

say they did not know and provide an unsure response. Students who made a bar graph with just a mean or mean and error bars were less likely to say they did not know. Students who made a bar graph with error bars were more likely to have a confident answer (Fig 8a). First year students were less likely to be unsure about their answer (Fig 8b). Specialized biology majors were more likely to have a confident answer and less likely to be unsure or say they did not know. Allied majors were also less likely to say they did not know. NonSTEM majors were more likely to say they did not know or were unsure and less likely to be confident in their answers (Fig 8c).

### Evidence for continually teaching graphing throughout biology curriculum

Academic year ($X^2$ = 92.6, df = 15, p < 0.0001) and major ($X^2$ = 55.98, df = 15, p < 0.0001) may play a factor in the type of graph created. First year students were more likely to make a raw bar graph and less likely to make a bar graph showing the mean or use error bars. Sophomores were more likely to make a bar graph with error bars and less likely to make a raw bar graph. Seniors were more likely to create an incorrect quantitative scatterplot (Fig 9a). Students who reported being general biology majors were more likely to make a raw bar graph and less likely to make an error bar graph. Specific biology majors were more likely to make an error bar graph. Allied majors were less likely to make a raw bar graph

**Table 3. Example quotes from students' responses to "What type of information do the error bars provide you about the data represented in each bar?" based on the graph they made.**

| Bar graph (raw) | Bar graph (ymean) | Bar graph (ymean + error bars) |
|---|---|---|
| "Error bars provide you with a visual representation of uncertainty in the measurements (margin of error)." R1, Intro, Junior, Allied | "The error bars tell the person interpreting the data that the data is "statistically significant" meaning that the data is not due to chance. The longer the error bars, the greater the standard deviation is from the mean. If the error bars overlap it is a cue that the data may not be statistically significant." R1, Intro, First year, GenBio | "The error bars indicate the highest data point and the lowest data point in relation to the site the lizard was found in. These error bars can indicate the range, or a calculated two standard deviation[s] of the data set. Error bars are generally used to provide insight into how precise a data set is." R2, Upper, Junior, SpecBio |
| "Errors bars provide an average quantity that shows the uncertainty in a calculated data, but sometimes provide the confidence interval of the average." R1, Intro, First year, GenBio | "The error bars represent how different the data within each category varied. Smaller error bars represent smaller variation meaning data is close together and results continue to yield similar numbers. Higher error bars means numbers were more varied and results yielded several vastly different numbers." R3, Intro, Sophomore, GenBio | "The error bars show the distribution of the data and gives us a better idea of the mean and overall data analysis." R1, Intro, Junior, GenBio |
| "The error bars represent potential errors in the study and indicate where the actual mean of the data may lie. The actual mean could be in any of the area indicated by the error bars. They essentially represent the uncertainty of the collected data." R1, Intro, First year, GenBio | "The error bars provide information about the accuracy of the data; the larger the bar, the more errors. If both bars align, the data is insignificant." R1, Intro, Sophomore, GenBio | "The error bars provide information about the amount of uncertainty in the data. Overlapping error bars indicate that the data is not statistically significant." R1, Intro, Sophomore, SpecBio |
| "The error bars provide variability. They represent the data as a whole, and what the spread of the data is like for each category." R1, Upper, Senior, GenBio | "The error bars provide you with standard deviations of the data that is plotted. Thus, the bars are telling you that there is data that lies outside of this average range. The greater the standard deviation, the greater the range of data that was obtained. The smaller the standard deviation, the smaller the range of data that was obtained." R1, Intro, Senior, GenBio | "The error bars provide better insight on the data, as it represents the standard deviation of values from each categorical study. Specifically, it shows how much values deviate from the mean, also representing [a] range of uncertainty." R1, Intro, First year, Allied |
| "Error bars represent a "window" of data points someone would likely get if they emulated the same experiment. The larger the error bars, the greater uncertainty there is surrounding the specific data point found." Masters, Nonmajor, Sophomore, NonSTEM | "The error bars show the potential error in the data and allow the reader to understand how low/high the average may actually be." R3, Upper, Senior, GenBio | "The error bars provide us with a data range that the mean falls in. When comparing the two graphs, bar overlapping can represent similarity." R1, Intro, First year, SpecBio |
| "The error bars help describe the uncertainty or variation of the graph. It describes [the] variability of the data. Using these can help show the margin of uncertainty based on the observed data." R1, Intro, First year, Allied | "The error bars represent the statistical uncertainty that comes with expressing the measured data. In this case, measuring lobster density may have left room for error (e.g., outlier in sample), and the error bars take such error into account and display the maximum density as a range." R3, Intro, First year, SpecBio | "The error bars measure the amount of variation/dispersion of the data set in comparison to the mean. The larger the variation in relation to mean and the more the outliers, the more spread out the standard deviation error bars are. Also, even though the average of the bar is higher for PF, because there is overlap, the standard deviation error bar lets us know that some or most of the values might be similar between the EC and PF, meaning there doesn't have to be a significant difference in the two datasets." R1, Intro, First year, Allied |

and more likely to make an error bar graph. NonSTEM majors were less likely to make an error bar graph (Fig 9b). Whether students were in an introductory or upper-level biology course did not have a significant impact on the type of graph that was made ($X^2 = 2.08$, df = 5, p = 0.838).

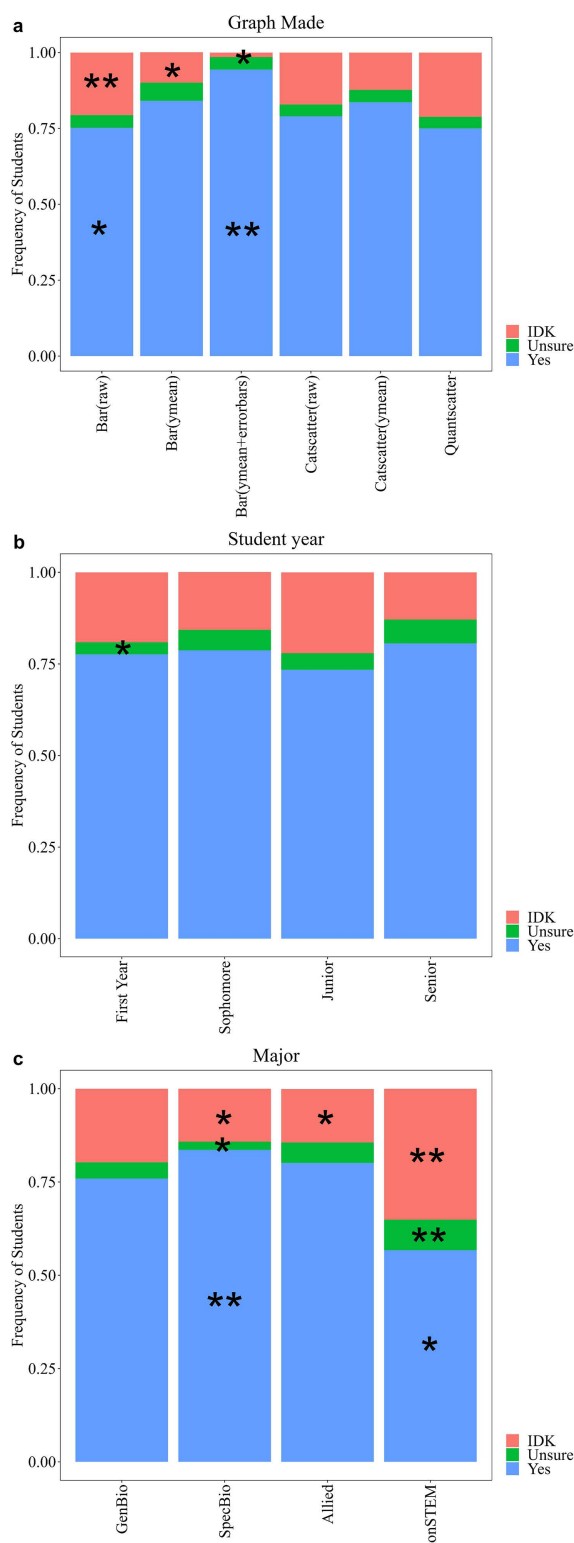

**Fig 8. Graphs students made by select demographics including student year (a), reported major (b), and course type (c).** A chi-square test was performed, and a single asterisk represents a frequency that was significantly lower than expected. Double asterisk represents a frequency that was significantly higher than expected. Frequencies do not add up to 100 for (a) or (b) due to not all students reporting demographics and for (c) due

not reporting on non-majors courses. Bar (raw) is a bar graph created with raw data; N = 1071. Bar (ymean) is a bar graph with just means; N = 484. Bar (ymean + errorbars) is a bar graph with means and error bars; N = 391. Catscatter (raw) is a graph similar to graph 3; N = 205. Catscatter (ymean) is a graph with a dot representing the mean for each group; N = 73. Quantscatter is a traditional scatterplot with two numerical variables; N = 184.

## Discussion

This study fills a gap in the current literature about what undergraduate biology students know about the data error bars represent or their ability to interpret variability across different graph types relevant to biology. Our thematic analysis of student responses revealed a code book showing *Broad Terms, Error Terms, Purpose,* and *Trend & Analysis* as four major categories for how students interpret error bars. Students demonstrated difficulty recognizing whether variability was present in a graph when asked to interpret data within a treatment. Students were more likely to say the variability was easier to interpret in a graph similar to a graph they created within our assessment. We saw a continuum of understanding regarding error bars according to the type of bar graph a student created. Students who graphed raw data on a bar graph showed less knowledge of error bars compared to students who graphed means and error bars. Students who graphed raw data on a bar graph were more likely to say they did not know what the error bar represented. We saw interesting trends within the demographics for student year and major suggesting more research is needed within STEM majors on how students are interpreting variability.

Students need more practice with reading variability in biologically relevant graph types. Our data supported results from other studies that showed students often view bar graphs as the best and easiest to read graph type [40,41]. Regardless of whether the bar graph was made with raw data or with means and error bars, students said the variability within these graphs was easy to interpret. We saw a large portion of students fail to identify when no variability was present on a bar graph, which may support a common misconception with bar graphs where students interpret the heights of the bars as being part of the variability [15,42]. Similarly, categorical scatterplots are not a graph type students commonly encounter within their classes (manuscript in prep), which could explain the lower proportion of students describing this style of graph as easy to interpret even though all points are visible. Few instructors regularly use categorical scatterplots across STEM discipline courses, though calls for their increased use have been made previously [43–45]. We did not observe noticeable trends with students who constructed categorical scatterplots like we did with students who made bar graphs.

Our dataset revealed a wide range of understanding for what error bars represent. Nineteen percent of students could not provide an answer to what error bars represented. This value is lower than studies with smaller sample sizes [46] such as Rahmatina et al. [47], where 59% could not perform reasoning about variability when interpreting graphs. Across our codes, we see many common definitions for error bars including error bars representing standard deviation, standard error, confidence intervals, and range [26,47]. Error bars inherently do show variability, so student responses given that codes are not inherently incorrect, but just saying that error bars show variability is a broad response. Other broad terms seen in the literature to describe variability include uncertainty and distribution [48,49].

Some student responses show incomplete understanding of error bars. Many students talked about the error bar overlapping or lack thereof providing information for the graph viewer. The overlap method is a quick, rough method for exploratory data analysis *if* the correct error bars are used with the graph, but the overlap method should not be used for formal significance testing [26,50]. Students describing error bars as being or showing outliers demonstrate a misconception. For example, one student said, "measuring lobster density may have left room for error (e.g., outlier in sample), and the error bars take such error into account," last example of bar graph (ymean), Table 3.) Research shows that participants reading graphs often don't disregard outliers and thus can have biased viewpoints [51,52]. We are limited to only what the students provided in their response, so following up on how error bars as outliers impacts students' interpretability of the graph would be interesting. Even when students used terms such as spread within their responses, Kaplan et al. [53] suggests that the term spread has diverse meanings and students might not be thinking

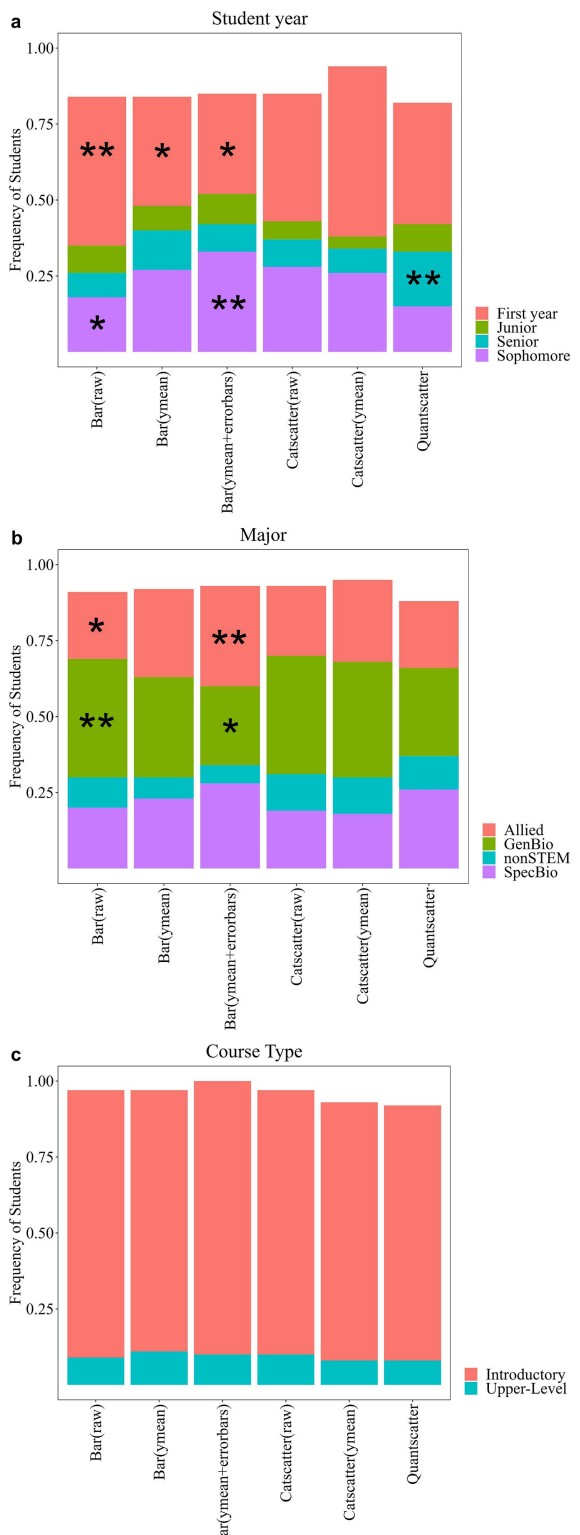

**Fig 9. Frequencies for answer confidence for students answering about what data are represented by error bars depending on what graph they made (a), what year they were (b), and what their reported major was (c).** A chi-square test was performed and a single asterisk represents a frequency that was significantly lower than expected. Double asterisk represents a frequency that was significantly higher than expected. Bar (raw)

is a bar graph created with raw data; N = 1071. Bar (ymean) is a bar graph with just means; N = 484. Bar (ymean + errorbars) is a bar graph with means and error bars; N = 391. Catscatter (raw) is a graph similar to graph 3; N = 205. Catscatter (ymean) is a graph with a dot representing the mean for each group; N = 73. Quantscatter is a traditional scatterplot with two numerical variables; N = 184.

of the statistical meaning within their response. Using terms including random, confidence, and association would provide less ambiguity in understanding student responses. Koklu & Kaplan [18] also suggest that codes such as distribution and range show less understanding of variability. Students also perceived variation as inherently negative when describing variation due to errors in data collection. This perception is untrue and variation is not inherently indicative of a mistake. McOsker [48] also saw a similar result with 66 undergraduate students that equated error in scientific papers with mistakes and inaccuracies.

To our surprise, only a handful of students recognized that we did not specify the type of error bar. While bar graphs and the use of error bars is common within STEM disciplines, students show little regard for understanding the different types of error bars. Descriptive error bars such as standard deviation and range cannot be used to estimate whether two treatments are different from one another like inferential error bars such as standard error and confidence intervals can [26]. Thus, inferential error bars should be used on graphs depicting biological data. When students create graphs that use error bars, instructors should make sure students understand the different error bar types, which are appropriate to use when, and how to use error bars to make predictions. Students should be learning basic statistical tests to compare sample means to back up their predictions when reading their graphs [53]. Belia et al. [27] also showed how researchers publishing in peer reviewed journals had trouble distinguishing between confidence intervals and standard error bars and how each can be used to interpret figures. So, while the problem is not only student based, we should be working to decrease the number of students and experienced researchers with misconceptions and misunderstandings regarding error bars.

We linked student answers with graphs created, which allowed us to see patterns in student reasoning that other studies have been unable to measure statistically. While students made a variety of graphs, we saw the most interesting trends between students who made raw bar graphs and students who made bar graphs with a mean and error bars. This result is not surprising since students often comment that bar graphs are "easier to read" and one of the most familiar graphs for students [40,41]. First year students were more likely to make a raw bar graph, which researchers such as Picone et al. [29] observed when 67% of students created a raw bar graph when interpreting ecological data. Gardner et al. [31] also noticed students tended to make raw bar graphs more often when hand drawing graphs compared to an online environment. Even though students encountered graphs in their K-12 years, many students enter college still thinking about visualizing individual data points instead of thinking of aggregating all data points within a sample [54].

Students who made bar graphs of raw data showed evidence for having a lower understanding of error bars because they used codes from *Broad Terms* only or *Error Terms* only. While some students who graphed raw bar graphs only used *Purpose* codes to explain error bars, their responses demonstrated one facet of potential understanding such as stating that the error bar showed outliers (Fig 7). The outlier code was one of the lower ranked purpose codes because while the size of error bars can change depending on if there are outliers in the data, the error bars themselves cannot tell an interpreter of the graph whether an outlier is present or absent. Participants tend to get distracted by outliers which can prevent them from seeing the overall trend when interpreting graphical information [29,51]. Students who made bar graphs with means and error bars showed a greater understanding of error bars and the data they represent. Students who made bar graphs with error bars used significantly more codes when answering the open-ended question, thus their responses more often included multiple statements. The codes within the *Purpose* and *Trend & Analysis* categories overall showed greater understanding than *Broad Terms* or *Error Terms* categories because of the greater specificity of what the codes represented. As mentioned above, codes within the *Purpose* and *Trend & Analysis* categories most closely align with definitions of error bars found in the literature.

The accuracy and precision code is interesting because we often think of accuracy and precision more within STEM disciplines such as physics and chemistry, where molecular parameters need to be specific and physics theories operate within specific parameters. Within biology, while you want to be accurate in conducting experiments, there is inherent biological variation that students often fail to take into consideration when interpreting their results. Thus, the measurements obtained from a biological study may not always have precision and may show a wider range than students are initially comfortable with. We see this in the literature when students are asked to interpret difficult graphical visualizations and often ignore certain parts of the graph, including structures of variability to make seeing the pattern more basic [15,40,47].

Our student population collectively did not show learning gains from their first year to final year in a program. We saw a significant increase from first year to sophomore students who chose to create a bar graph with error bars compared to raw bar graphs, but this trend did not continue with juniors and seniors. One of the more interesting non-significant results showed no difference in how students made graphs based on whether they reported being in an introductory course or an upper-level course. We would have expected students to show growth with more upper-level course students choosing to construct the more appropriate bar graph with error bars. This increase in understanding was observed within K-12 students, where students in higher grades tended to give more intricate answers for their understanding of variability, which reflected a greater level of understanding [55–57]. When assessing the impact of graphing instruction and active learning approaches on the ability to interpret and construct graphs within an upper division biology course, Weigel & Angra [58] noticed students may ignore variability when interpreting their findings. Students additionally struggled with visual representations such as standard deviation and confidence intervals and how those variability features are used in the interpretation of graphs.

Quantitative reasoning skills are used across all STEM and non-STEM disciplines. Even the general public should confidently read graphs and understand the basic notions of variability. We saw interesting differences among students' self-reported majors. Because non-STEM students don't encounter as much graphing related content as STEM students, we were unsurprised to see that non-STEM students were less likely to graph with error bars and more likely to say they did not know for the open-ended question. These results stem from a general unfamiliarity with the features of graphs including error bars. Elrod & Park [46] had both STEM majors and non-STEM majors complete the Quantitative Literacy and Reasoning Assessment within their math courses and reported a 5.6% increase in STEM major scores compared with non-STEM majors. Meletiou-Mavrotheris & Lee [59] showed that non-STEM majors entering an introductory statistics course exhibited difficulties in graphing tasks including graph construction and interpretation. While studies commonly differentiate between STEM and non-STEM students, differentiation of specific STEM majors is uncommon within the literature. One study explored how disciplinary knowledge for biology and chemistry students aided in interpretation of scientific graphs [59]. The researchers saw students who did not employ specific disciplinary knowledge often had lower levels of reasoning compared to students who did employ specific disciplinary knowledge. They also observed students who had high levels of reasoning without using specific disciplinary knowledge. Students from allied majors such as chemistry and engineering may see more graph related content compared to biology students, resulting in their ability to better apply knowledge about graphs. Further exploration is needed to understand potential differences across STEM majors in how their graphing knowledge.

## Conclusions

Students have wide and often incomplete understanding of variability when interpreting biological data. Our results showed many interesting ways students across institution types, course levels, and year explain variability and the data error bars represent. We echo several other studies [7,8,28,41,60,47], which suggest scaffolding variability within the appropriate context is important for students to continuously get experience with variability concepts. One way to easily incorporate biological context into scaffolding of variability is through using 'messy data', which gives students a chance to work with real data and observe that natural biological variation [7,8]. In courses with quantitative components, assessing students' prior knowledge of quantitative reasoning could identify areas where students excel or struggle. This information could assist the instructor in where to devote class time for strengthening these skills [61,62]. We suggest making sure

students understand the different types and uses of error bars if you want them to construct bar graphs showing aggregated data. Exposure to other graphs that show variability such as categorical scatterplots and even boxplots with raw data will expose students to additional ways to visually view variability. Repetition of graphing and the facets of variability will allow students to retain this information and better prepare our future STEM professionals.

## Supporting information

**S1 File. Appendix A- Chi square residuals.**
(XLSX)

**S2 File. Appendix B- Variability codebook.**
(DOCX)

**S3 File. R code.**
(DOCX)

**S4 File. FirstClassVariabilityData.**
(XLSX)

**S5 File. Variability by graphs.**
(XLSX)

## Acknowledgments

We would like to thank all the instructors that used our GraphSmarts assessments in their classrooms to collect such an extensive dataset.

## Author contributions

**Conceptualization:** Lauren Stoczynski, Stephanie M. Gardner.

**Data curation:** Susan Maruca, Eli Meir.

**Formal analysis:** Lauren Stoczynski, David Zis, Anna Woodruff.

**Funding acquisition:** Eli Meir, Joel K. Abraham, Stephanie M. Gardner.

**Methodology:** Lauren Stoczynski, Stephanie M. Gardner.

**Project administration:** Stephanie M. Gardner.

**Software:** Lauren Stoczynski, Susan Maruca, Eli Meir.

**Supervision:** Stephanie M. Gardner.

**Visualization:** Lauren Stoczynski.

**Writing – original draft:** Lauren Stoczynski.

**Writing – review & editing:** David Zis, Anna Woodruff, Susan Maruca, Eli Meir, Joel K. Abraham, Stephanie M. Gardner.

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
