## [Decision Letter · Decision Letter 0]

4 Dec 2025

Dear Dr. Stoczynski,

Thank you for submitting your manuscript to PLOS ONE. After careful consideration, we feel that it has merit but does not fully meet PLOS ONE’s publication criteria as it currently stands. Therefore, we invite you to submit a revised version of the manuscript that addresses the points raised during the review process.

**The study is impressively large and your analysis is quite thorough, as both reviewers**
**noted. Your descriptions of the value of this study also should be commended. Despite quantitative literacy being a pillar of Vision and Change, we too often do not systematically assess our students knowledge of graphing, error bars, statistical significance, etc. This work help fill this gap.****One reviewer provided many methodological and data presentation suggestions. I would encourage you to review these points. In my opinion though, these should just be considered as suggestions. The other reviewer raised one major concern regarding the figures. Please check Fig. 8 and Fig. 9. Both are mentioned in the text, but it appears that one of them is not attached to the submission.**

We look forward to receiving your revised manuscript.

Kind regards,

David R Wessner, Ph.D.

Academic Editor

PLOS ONE

“This research received grants from the National Science Foundation under grant Nos 1726180 and 2111150. Any opinions, findings, and conclusions or recommendations expressed in this material are those of the author(s) and do not necessarily reflect the views of the National Science Foundation.”

4. We note that there is identifying data in the Supporting Information file <FirstClassVariabilityData.xlsx and variability by graphs.xlsx>. Due to the inclusion of these potentially identifying data, we have removed this file from your file inventory. Prior to sharing human research participant data, authors should consult with an ethics committee to ensure data are shared in accordance with participant consent and all applicable local laws.

-Location data

Please remove or anonymize all personal information (ID numbers), ensure that the data shared are in accordance with participant consent, and re-upload a fully anonymized data set. Please note that spreadsheet columns with personal information must be removed and not hidden as all hidden columns will appear in the published file.

Reviewers' comments:

Reviewer's Responses to Questions

**Comments to the Author**

1. Is the manuscript technically sound, and do the data support the conclusions?

Reviewer #1: Partly

Reviewer #2: Yes

2. Has the statistical analysis been performed appropriately and rigorously?

Reviewer #1: No

Reviewer #2: Yes

3. Have the authors made all data underlying the findings in their manuscript fully available?

Reviewer #1: Yes

Reviewer #2: Yes

4. Is the manuscript presented in an intelligible fashion and written in standard English?

Reviewer #1: Yes

Reviewer #2: Yes

Reviewer #1: 1. The study presents the results of original research

This study appears to present the results of original research. The topic is timely, and the authors demonstrate technical proficiency in both quantitative and qualitative methods.

2. Results have not been published elsewhere

It does not appear that the results have been reported elsewhere. The work seems to be novel and contributes original findings to the field.

3. Experiments, statistics, and analyses

a. Statistical Analysis

The study employs over twenty chi-square tests on a large sample and produces more than twenty plots to illustrate category-level percentages. I recommend that the authors consider a more efficient and integrative approach. Rather than conducting a series of separate chi-square tests with a common dependent variable (for example, “Made Raw Bar Graphs”), the authors could model all outcomes simultaneously using multinomial logistic regression. If the dependent variable were Type of Graph Created (e.g., “Raw Bar,” “Error Bar,” “Mean Graph”), the predictors such as interpreting raw bar graphs or interpreting error bar graphs could be entered into a single multinomial logistic regression model. This would allow the researchers to estimate the probability of each outcome while controlling for all predictors in a single model rather than running multiple independent tests.

There are several limitations to running multiple chi-square tests. Large sample sizes make chi-square tests prone to significance even when the practical differences are negligible. Moreover, conducting multiple binary tests with one independent and one dependent variable increases the likelihood of a Type I error or false positive. In contrast, multinomial logistic regression offers several advantages. It reduces familywise error and thus decreases the likelihood of Type I error. It provides a more efficient analytic framework by integrating predictors into one model, and it produces interpretable coefficients and odds ratios that facilitate comparison across predictors. Finally, it supports concise tabular presentation of results, which is considerably more reader-friendly than numerous bar plots. Overall, this approach would yield more coherent and statistically robust conclusions that are well supported by the data.

I encourage the authors to consider this method using the multinom() function from the nnet package in R, which performs multinomial logistic regression. This approach allows multiple categorical outcomes to be modeled simultaneously rather than conducting a series of separate chi-square tests. The multinom() function efficiently estimates the probability of each outcome category as a function of one or more predictors, offering odds ratios and confidence intervals that enhance interpretability and parsimony.

For example, if the dependent variable were Graph_Type (e.g., "raw", "error_bar", "mean") and the independent variables were Interpreted_Error_Bar, Interpreted_Raw_Bar, and Interpreted_Mean_Bar, a simple model could be specified as:

library(nnet)

model <- multinom(Graph_Type ~ Interpreted_Error_Bar + Interpreted_Raw_Bar + Interpreted_Mean_Bar, data = your_data)

summary(model)

This example is generic. I encourage the authors to consider whether this would work and research it and to feel free to dispute this recommendation in their response if it does not work, explaining why.

Be sure to include the regression tables in your manuscript. These would include the coefficients for each predictor relative to the reference category of the dependent variable, the standard errors, z-values, p-values, and odds ratios with 95% confidence intervals. It can also be helpful to include predicted probabilities for each outcome category to illustrate the practical interpretation of the model. Organizing the table so that each dependent variable category is clearly labeled and showing all relevant predictors side by side will make it easier for readers to understand the results without having to refer to multiple figures.

b. Multiple Figures vs. One Table

As an R user, I genuinely appreciate the researchers’ figures. They demonstrate a strong command of R’s advanced graphing capabilities and reflect thoughtful attention to data visualization. However, I recommend reconsidering the presentation format. The inclusion of more than twenty figures, each depicting a subset of categorical comparisons, makes the results difficult to follow and interpret as a coherent whole. A more effective alternative would be to replace the numerous figures with a single comprehensive table that displays the counts and percentages for all relevant variable combinations, with clear labeling to indicate which variables or categories each row represents.

A well-designed table would allow readers to view all comparisons at once, identify patterns and relationships across variables more efficiently, reference specific values precisely, and cross-validate their interpretation against the statistical results, such as those produced by a multinomial logistic regression. If the authors wish to retain visualization for accessibility and readability, they might consider supplementing the table with one or two summary figures that highlight key findings or effect sizes, such as predicted probabilities derived from the regression model. This would balance visual appeal with analytic clarity. The table itself should be self-contained, including variable names, category labels, sample sizes, and proportions, and it should mirror the structure of the statistical model. Each dependent variable category could be presented as a block, with independent variables shown alongside the relevant counts, percentages, and model-derived statistics such as odds ratios, confidence intervals, and p-values. Adopting this approach would substantially enhance clarity, interpretability, and reproducibility while preserving the authors’ evident technical skill in R.

c. Qualitative Portion

The qualitative section of the study is interesting, particularly in its use of coded data as numeric variables for some of the chi-square tests. There are, however, several areas that could be strengthened. First, I recommend consulting and citing Saldaña’s Coding Manual for Qualitative Researchers (2025 edition). This resource distinguishes between “thematic coding” and “theming the data,” which may help the authors refine their methodological description.

Second, greater clarity is needed regarding the term “inter-rater reliability.” The text suggests that this might refer instead to inter-coder agreement, where a given percentage of codes overlap between coders. Inter-rater reliability typically refers to consistency among raters using a psychometric instrument such as a rubric, whereas inter-coder agreement assesses similarity in qualitative coding. The authors should clarify their intended meaning and provide appropriate citations.

Finally, more detail on the coding process would strengthen this section. The authors mention both inductive coding and the use of a codebook, which suggests a hybrid or sequential approach. Inductive coding generally implies open coding without predefined categories, while a codebook implies a deductive or a priori framework. Clarifying whether the codebook was generated after an initial inductive phase and subsequently applied deductively would help readers better understand the analytic process.

d. Citations of Methods

I also recommend citing and referencing all R packages and software used in the analysis, as well as any qualitative sources that informed the methodology. Within the R community, it is considered both best practice and professional courtesy to credit package developers, particularly in publications such as PLOS that value open science and reproducibility. Including these citations would also enhance the transparency and replicability of the study and will help the developers of the code.

4. Research Ethics

I believe strong research ethics and publication standards were followed in this study. The research appears to meet all applicable standards for the ethics of experimentation and research integrity.

5. Reporting and Data Availability

The article generally adheres to appropriate reporting guidelines and community standards of data availability. Sharing the R code used in the analysis would further strengthen transparency and reproducibility and would help readers understand how the models were derived from the multiple-tab Excel file.

6. Final Thoughts

Overall, this is a promising and well-conceived study that demonstrates both technical and methodological sophistication. My primary recommendations are to consolidate the statistical analyses through multinomial logistic regression, simplify and clarify result presentation by replacing numerous figures with a comprehensive table, and provide additional methodological detail in the qualitative section. Incorporating citations for software and methodological sources, along with sharing the analytic code, would further enhance the transparency, rigor, and overall contribution of the paper.

Reviewer #2: This paper reports findings from an impressively large study of undergraduate biology students interpreting graphs, focussing on their understanding of error bars and data variability. The results are interesting. There are some reasonably expected results, showing for example how students who generate plots with error bars have better understanding of them, people tending to find the type of plot they make as easier to understand, and reasonable shifts in descriptions based on whether they made raw bar graphs or those with error bars. Counterintuitive results included no apparent increase in understanding or confidence with increasing year of study. This work highlights important features of teaching plotting data to biology students and importance of conveying meaning and use of error bars.

Overall the study is robust and well done, and the report is clearly written. There are a few areas that need addressing:

Major comments:

1. There is a figure missing. The legend for Fig 9 matches what is provided as Fig 8; there is no figure provided that matches the description of Fig 8. Please provide the missing figure.

2. Throughout, it is noted if things are “significant” but the methods do not provide information about the significance criteria applied. Please add a line to the “Quantitative data analysis” section indicating what criteria (e.g., P<0.05, P<0.01) is used. Additionally, it is unclear if any correction for multiple testing is applied before evaluating significance from the chi-square/Fisher’s or Kruskal Wallis tests. Please indicate if any correction to the p-value or significant criteria were applied to correct for multiple tests and justify what is considered together in one unit for the ‘multiple’; if none were applied, please justify this.

3. Please provide information in the methods as to what you assessed as the correct answer to the “easy to interpret”/“hard to interpret”/“not shown” question, and explanation for how you determined what is easy vs hard (it looks to me like either error bars or individual values provided in an ordered matter made it “easy”).

4. I worry that some of the axis labels and in-figure legends will be unreadably small in a PDF/print version. Only Fig 1 and 4 look reasonably sized – please increase text size in others.

5. Line 848: it appears the legend for Fig 5 is incomplete. Please add the information alluded to about the number of each graph type the results are based on. Also please explain the terms used to describe each type of graph generated by students, e.g., Bar(raw) = value bar chart, etc, (in particular I’m not sure what two cat ones are? I’m thinking ‘bar and whiskers’ box plot types, but please explain.)

6. Lines 885-886: I am struggling to match up the letters A-F with significant post-hoc comparisons – please clarify what each letter refers to for the plots.

Minor comments:

1. Lines 444-446: This sentence in the abstract, “When responses were linked with a graph the student made, those who created a bar graph with raw data showed lower understanding of error bars compared to students who themselves created a bar graph with aggregated data and error bars.”, is slightly confusing without context as it is not clear what ‘raw’ vs ‘aggregated’ data means. I suggest something a little more like “a bar graph showing every data point as a separate bar” vs “a bar graph showing aggregated means and error bars”

2. Lines 471 & 472: something is cited for 10 competencies, but they are not named, so the reader has to back-think to this sentence when “plan competency” and “analyze competency” is mentioned. I suggest either naming the 10, or say something like “including plan and analyze”.

3. Line 480: no comma before (2020)

4. Lines 501-502: I’m not sure ‘value’ instead of a ‘mean’ is that clear an explanation of a value bar chart, since a mean is also a type of value. Potentially something like “an individual data point’s value” would be clearer.

5. Lines 570-571: misplaced newline after “whether”

6. Line 578: no apostrophe in “question’s” (should be “questions”)

7. Line 601 (Table 1 legend), 668 (Fig 3 legend), 800 (Fig 4 legend): these all mention “the first time” for the students taking the assessment, however, it is already specified in methods that all the analysis is about the first time students take this – leaving this in the legends implies you are going to analyse more, so I suggest removing it from these legends.

8. Line 671: elsewhere it says “Error Terms” instead of “Form of Error”, so I suggest using “Error Terms” here as well, or just “Error”

9. Lines 676-765: please refer to appropriate sections of Figure 3 when reporting the relevant data

10. Line 985: there is an extraneous closing parenthesis at the end of this line.

11. Line 990: no comma before (2009)

12. Line 1010: no comma before (2005)

13. Line 1028: “thinking of the sample size as something that should be aggregated” is a little confusing – it’s not the sample size that is being aggregated but the individual values for each sample in the sample size. Perhaps something more like “thinking of aggregating across all data points within a sample”

**Do you want your identity to be public for this peer review?** For information about this choice, including consent withdrawal, please see our Privacy Policy

Reviewer #1: **Yes:** Mark A. Perkins

Reviewer #2: No

---

## [Author Response · Author response to Decision Letter 1]

20 Jan 2026

Due to inclusion of potentially identifying data, the excel files were removed from the file inventory. We removed the “CourseID” column as that was the only column I could see that they were saying was identifying from FirstClassVariabilityData file. From the variability by graphs file, removed the timestamp and courseID columns.

Reviewers' comments:

Reviewer's Responses to Questions

Comments to the Author

1. Is the manuscript technically sound, and do the data support the conclusions?

Reviewer #1: Partly

Reviewer #2: Yes

2. Has the statistical analysis been performed appropriately and rigorously?

Reviewer #1: No

Reviewer #2: Yes

3. Have the authors made all data underlying the findings in their manuscript fully available?

Reviewer #1: Yes

Reviewer #2: Yes

4. Is the manuscript presented in an intelligible fashion and written in standard English?

Reviewer #1: Yes

Reviewer #2: Yes

5. Review Comments to the Author

Reviewer #1: 1. The study presents the results of original research

This study appears to present the results of original research. The topic is timely, and the authors demonstrate technical proficiency in both quantitative and qualitative methods.

2. Results have not been published elsewhere

It does not appear that the results have been reported elsewhere. The work seems to be novel and contributes original findings to the field.

3. Experiments, statistics, and analyses

a. Statistical Analysis

The study employs over twenty chi-square tests on a large sample and produces more than twenty plots to illustrate category-level percentages. I recommend that the authors consider a more efficient and integrative approach. Rather than conducting a series of separate chi-square tests with a common dependent variable (for example, “Made Raw Bar Graphs”), the authors could model all outcomes simultaneously using multinomial logistic regression. If the dependent variable were Type of Graph Created (e.g., “Raw Bar,” “Error Bar,” “Mean Graph”), the predictors such as interpreting raw bar graphs or interpreting error bar graphs could be entered into a single multinomial logistic regression model. This would allow the researchers to estimate the probability of each outcome while controlling for all predictors in a single model rather than running multiple independent tests.

There are several limitations to running multiple chi-square tests. Large sample sizes make chi-square tests prone to significance even when the practical differences are negligible. Moreover, conducting multiple binary tests with one independent and one dependent variable increases the likelihood of a Type I error or false positive. In contrast, multinomial logistic regression offers several advantages. It reduces familywise error and thus decreases the likelihood of Type I error. It provides a more efficient analytic framework by integrating predictors into one model, and it produces interpretable coefficients and odds ratios that facilitate comparison across predictors. Finally, it supports concise tabular presentation of results, which is considerably more reader-friendly than numerous bar plots. Overall, this approach would yield more coherent and statistically robust conclusions that are well supported by the data.

I encourage the authors to consider this method using the multinom() function from the nnet package in R, which performs multinomial logistic regression. This approach allows multiple categorical outcomes to be modeled simultaneously rather than conducting a series of separate chi-square tests. The multinom() function efficiently estimates the probability of each outcome category as a function of one or more predictors, offering odds ratios and confidence intervals that enhance interpretability and parsimony.

For example, if the dependent variable were Graph_Type (e.g., "raw", "error_bar", "mean") and the independent variables were Interpreted_Error_Bar, Interpreted_Raw_Bar, and Interpreted_Mean_Bar, a simple model could be specified as:

library(nnet)

model <- multinom(Graph_Type ~ Interpreted_Error_Bar + Interpreted_Raw_Bar + Interpreted_Mean_Bar, data = your_data)

summary(model)

This example is generic. I encourage the authors to consider whether this would work and research it and to feel free to dispute this recommendation in their response if it does not work, explaining why.

Be sure to include the regression tables in your manuscript. These would include the coefficients for each predictor relative to the reference category of the dependent variable, the standard errors, z-values, p-values, and odds ratios with 95% confidence intervals. It can also be helpful to include predicted probabilities for each outcome category to illustrate the practical interpretation of the model. Organizing the table so that each dependent variable category is clearly labeled and showing all relevant predictors side by side will make it easier for readers to understand the results without having to refer to multiple figures.

While we appreciate the reviewer’s comments in suggesting using a multinomial logistic regression, upon doing some research and using the code the reviewer provided, we have decided to stay with our chi square approach. The multinomial logistic regression pulled out one of the independent variables and compared the other independent variables to that one. This approach would have meant to get the full model, we would have needed to run multiple multinomial logistic regressions designating a different reference variable. We also still would have needed multiple regressions for each of the different questions we asked, which was a similar vein to the chi-square test. We have applied a post hoc analysis (Bonferroni correction) to the chi-square tests to reduce the probability of a Type I error, which caused some minor changes to our results. In the supplemental material we will provide the critical value and residual table for each of the chi-square tests.

b. Multiple Figures vs. One Table

As an R user, I genuinely appreciate the researchers’ figures. They demonstrate a strong command of R’s advanced graphing capabilities and reflect thoughtful attention to data visualization. However, I recommend reconsidering the presentation format. The inclusion of more than twenty figures, each depicting a subset of categorical comparisons, makes the results difficult to follow and interpret as a coherent whole. A more effective alternative would be to replace the numerous figures with a single comprehensive table that displays the counts and percentages for all relevant variable combinations, with clear labeling to indicate which variables or categories each row represents.

While we appreciate the reviewer's suggestion, we think the data are easier to visualize within figures and not within a comprehensive table. Several of the figures have been reformatted for better readability and we thank the reviewer for encouraging us to reflect on how to best present our data.

A well-designed table would allow readers to view all comparisons at once, identify patterns and relationships across variables more efficiently, reference specific values precisely, and cross-validate their interpretation against the statistical results, such as those produced by a multinomial logistic regression. If the authors wish to retain visualization for accessibility and readability, they might consider supplementing the table with one or two summary figures that highlight key findings or effect sizes, such as predicted probabilities derived from the regression model. This would balance visual appeal with analytic clarity. The table itself should be self-contained, including variable names, category labels, sample sizes, and proportions, and it should mirror the structure of the statistical model. Each dependent variable category could be presented as a block, with independent variables shown alongside the relevant counts, percentages, and model-derived statistics such as odds ratios, confidence intervals, and p-values. Adopting this approach would substantially enhance clarity, interpretability, and reproducibility while preserving the authors’ evident technical skill in R.

c. Qualitative Portion

The qualitative section of the study is interesting, particularly in its use of coded data as numeric variables for some of the chi-square tests. There are, however, several areas that could be strengthened. First, I recommend consulting and citing Saldaña’s Coding Manual for Qualitative Researchers (2025 edition). This resource distinguishes between “thematic coding” and “theming the data,” which may help the authors refine their methodological description.

Second, greater clarity is needed regarding the term “inter-rater reliability.” The text suggests that this might refer instead to inter-coder agreement, where a given percentage of codes overlap between coders. Inter-rater reliability typically refers to consistency among raters using a psychometric instrument such as a rubric, whereas inter-coder agreement assesses similarity in qualitative coding. The authors should clarify their intended meaning and provide appropriate citations.

Finally, more detail on the coding process would strengthen this section. The authors mention both inductive coding and the use of a codebook, which suggests a hybrid or sequential approach. Inductive coding generally implies open coding without predefined categories, while a codebook implies a deductive or a priori framework. Clarifying whether the codebook was generated after an initial inductive phase and subsequently applied deductively would help readers better understand the analytic process.

Thank you for the feedback and suggestions for strengthening this section of the manuscript. We appreciate the clarity that your suggestions will provide. We have referenced Saldaña and others to describe our qualitative data analysis approach but have added more detail. As noted by the reviewer, inter-coder agreement is aligned with our approach and incorporation of qualitative data to answer our research questions and complement the quantitative analysis. We have revised that section of the manuscript to reflect that. We have additionally revised our description of the step-wise process we applied with respect to how the codebook was generated and used to analyze our qualitative data.

d. Citations of Methods

I also recommend citing and referencing all R packages and software used in the analysis, as well as any qualitative sources that informed the methodology. Within the R community, it is considered both best practice and professional courtesy to credit package developers, particularly in publications such as PLOS that value open science and reproducibility. Including these citations would also enhance the transparency and replicability of the study and will help the developers of the code.

We appreciate the reminder from the reviewer to cite R and the FSA package. These citations have been added to the manuscript.

4. Research Ethics

I believe strong research ethics and publication standards were followed in this study. The research appears to meet all applicable standards for the ethics of experimentation and research integrity.

5. Reporting and Data Availability

The article generally adheres to appropriate reporting guidelines and community standards of data availability. Sharing the R code used in the analysis would further strengthen transparency and reproducibility and would help readers understand how the models were derived from the multiple-tab Excel file.

We will provide the R code used in these analyses with the supplemental information.

6. Final Thoughts

Overall, this is a promising and well-conceived study that demonstrates both technical and methodological sophistication. My primary recommendations are to consolidate the statistical analyses through multinomial logistic regression, simplify and clarify result presentation by replacing numerous figures with a comprehensive table, and provide additional methodological detail in the qualitative section. Incorporating citations for software and methodological sources, along with sharing the analytic code, would further enhance the transparency, rigor, and overall contribution of the paper.

Reviewer #2: This paper reports findings from an impressively large study of undergraduate biology students interpreting graphs, focusing on their understanding of error bars and data variability. The results are interesting. There are some reasonably expected results, showing for example how students who generate plots with error bars have better understanding of them, people tending to find the type of plot they make as easier to understand, and reasonable shifts in descriptions based on whether they made raw bar graphs or those with error bars. Counterintuitive results included no apparent increase in understanding or confidence with increasing year of study. This work highlights important features of teaching plotting data to biology students and importance of conveying meaning and use of error bars.

Overall the study is robust and well done, and the report is clearly written. There are a few areas that need addressing:

Major comments:

1. There is a figure missing. The legend for Fig 9 matches what is provided as Fig 8; there is no figure provided that matches the description of Fig 8. Please provide the missing figure.

We appreciate the reviewer pointing this out, Figure 9 is now included with the manuscript

2. Throughout, it is noted if things are “significant” but the methods do not provide information about the significance criteria applied. Please add a line to the “Quantitative data analysis” section indicating what criteria (e.g., P<0.05, P<0.01) is used. Additionally, it is unclear if any correction for multiple testing is applied before evaluating significance from the chi-square/Fisher’s or Kruskal Wallis tests. Please indicate if any correction to the p-value or significant criteria were applied to correct for multiple tests and justify what is considered together in one unit for the ‘multiple’; if none were applied, please justify this.

We have noted that the significance threshold is 0.05 on line 642. Additionally, we have noted that a Bonferroni correction was used with all analyses to reduce the probability of Type I error on line 652.

3. Please provide information in the methods as to what you assessed as the correct answer to the “easy to interpret”/“hard to interpret”/“not shown” question, and explanation for how you determined what is easy vs hard (it looks to me like either error bars or individual values prov

---

## [Editor Report · Decision Letter 1]

5 Feb 2026

Revealing undergraduate biology students' conception of variability and error bars within graphing

PONE-D-25-48061R1

Dear Dr. Stoczynski,

We’re pleased to inform you that your manuscript has been judged scientifically suitable for publication and will be formally accepted for publication once it meets all outstanding technical requirements.

Kind regards,

David R Wessner, Ph.D.

Academic Editor

PLOS One

Additional Editor Comments (optional):

Thank you for submitting this revised manuscript and for thoroughly and thoughtfully considering the suggests from the reviewers. Although both reviewers had several questions about the original manuscript, they also both noted that it described an impressive body of work and should contribute meaningfully to the literature. With the changes you made in response to their comments, the revised manuscript is much stronger. It should be of general interest to STEM educators.

Thanks for submitting this work to PLOS One.
---

## [Editor Report · Acceptance letter]

PONE-D-25-48061R1

PLOS One

Dear Dr. Stoczynski,

I'm pleased to inform you that your manuscript has been deemed suitable for publication in PLOS One. Congratulations! Your manuscript is now being handed over to our production team.

Kind regards,

on behalf of

Dr. David R Wessner

Academic Editor

PLOS One